# Enterprise Financialization and Technological Innovation: An Empirical Study Based on A-Share Listed Companies Quoted on Shanghai and Shenzhen Stock Exchange

**Tao Zhu and Xinyu Sun \***

School of Economics, Henan University, Kaifeng 475000, China
* Correspondence: 104753210111@henu.edu.cn

**Abstract:** In recent years, the growth rate of China's real industry has slowed down while the financial industry has entered a phase of rapid development. Driven by the profit-seeking motive of capital, real enterprises tend to carry out financial investments, and the degree of corporate financialization has been rising. This paper selects A-share listed enterprises in Shanghai and Shenzhen from 2009 to 2020 as research samples to study the impact of corporate financialization on technological innovation and the mediating effect of financing constraints from the perspective of financial asset holding. The study found that the financialization of enterprises' crowding out effect on technological innovation has led to the phenomenon of "turning from real to virtual". We also found that the crowding-out effect had experienced lag. This conclusion still held when we controlled for endogeneity. The heterogeneity analysis showed that the financialization of non-state-owned enterprises had an excessive inhibitory effect on technological innovation, and the financialization of enterprises in eastern China has had a remarkable inhibitory effect on technological innovation. The influence mechanism analysis showed how financing constraints played a crucial mediating role in corporate financialization inhibiting technological innovation, and corporate financialization has inhibited technological innovation by exacerbating financing constraints. Based on this research, we propose targeted suggestions to prevent the excessive financialization of enterprises on both government and enterprise levels.

**Keywords:** corporate financialization; technological innovation; financing constraints; asset allocation

## 1. Introduction

China's real economy is currently in the process of transformation and upgrading, while innovation has become the crucial driving force of national economic development. Enterprises have successively enhanced their innovation consciousness and technology innovation capability. However, China's traditional economic growth is faced with problems such as overcapacity, high investment costs, and a lack of core technologies. These problems have led to a repeated reduction in returns on investment in the manufacturing industry and declining market demand. Compared with the traditional manufacturing industry, it has become an indisputable fact that the rapid development of the financial industry could create an excessive profit rate. The income of financial investment made by non-financial enterprises exceeds the income of entity investment. Driven by the profit-seeking motive of capital, industrial capital has been withdrawn from the real economy and begun to continuously pour into the financial industry, bringing higher yields, which could lead to the uneven distribution of corporate assets, ignoring the development of main businesses, accelerating the expansion of the virtual economy, and eventually resulting in the phenomenon of "from the real to the virtual" [1]. According to the statistics of the CSMAR database, the financial assets held by Chinese real enterprises were approximately 1.04 billion RMB in 2008, while the financial assets held by Chinese real enterprises reached 3.28 billion RMB in 2020 [2]. The financialization of enterprises has a crucial impact on

enterprise innovation activities and brings a mass of challenges to the innovation and development of real enterprises. From a macro perspective, the economy is the body, while finance is the bloodline. Technological innovation and the high-quality development of the real economy need the "reservoir effect" of financial assets. From a micro perspective, the excessive financialization of enterprises affects the overall uneven distribution of resources, which produces a "crowding out effect" on the technological innovation of enterprises.

The report of the 19th National Congress of the Communist Party of China clearly stated that it is indispensable to "deepen the reform of the financial system, enhance the ability of financial services to the real economy, and guard the bottom line of not occurring systemic financial risks". At present, the Chinese economy has shifted from a stage of high-speed growth to a stage of high-quality development. Preventing the real economy from pursuing real development is the foundation for high-quality economic development. The Fifth Plenary Session of the 19th Central Committee of the Communist Party of China pointed out that it is imperative to "adhere to the core position of innovation in the overall situation of modernization, improve the technological innovation ability of enterprises, and accelerate the construction of a prosperous country in science and technology". Scientific and technological innovation accelerates the transformation of the economy from "quantitative development" to "qualitative development", which plays a crucial role in the transformation of an economic development mode. Therefore, the country is supposed to put scientific and technological innovation in the core position of overall national development. The 14th Five-Year Plan emphasized how "the government ought to maintain the proportion of the manufacturing sector as basically stable and consolidate and strengthen the foundation of the real economy". Promoting and strengthening the development of the real economy has been an indispensable task in China's economic construction since entering the new era. Scientific and technological innovation is the engine for real enterprises to achieve high-quality economic development. These policy directions show that the country attaches great importance to enterprise financialization and enterprise technological innovation.

Corporate financialization and technological innovation have always been popular research areas in the field of corporate finance. Their research value is ponderable. The purpose of this paper is to explore whether the financialization of real enterprises has a "reservoir effect" or "crowding out effect" for the financialization of enterprise technological innovation and whether the phenomenon of enterprise financialization can alleviate the financing constraint of technological innovation or intensify the financing constraint of technological innovation. Through the research of this paper, first of all, we aim to deepen an understanding of enterprise financialization at the micro level, explore the financial reasons for a lack of internal innovation power of enterprises, and correctly understand the current economic boom of "moving from the real to the virtual". Secondly, we clarify the intermediary role of financing constraints, promote financial financing efficiency, and strengthen their innovation input. Third, we aim to guide enterprises to rationally allocate financial assets and prevent the negative impact of excessive financialization. Fourthly, we provide ideas for the government to formulate macroeconomic policies and achieve high-quality economic development.

The subsequent content of this paper is arranged as follows. The second part is the literature review. The third part includes the theoretical analysis and research hypotheses. The fourth part is the research design, including sample selection, variable selection, and a description and benchmark model construction. The fifth part is the empirical analysis, including descriptive statistical analysis, benchmark model regression analysis, heterogeneity analysis, impact mechanism analysis, and robustness test. The sixth part details the conclusions and policy recommendations.

## 2. The Literature Review

The concept of enterprise financialization was put forward in the 1990s. Domestic and foreign scholars have studied much on the issue of enterprise financialization. Foreign scholars have defined financialization from both macro and micro perspectives. From a macro perspective, Palley (2007) pointed out that financialization refers to the process in which the proportion of financial markets, financial institutions, and financial activity participate in an economy that is gradually increasing [3]; From a micro perspective, Krippner (2005) believed that enterprise financialization referred to the asset allocation of entity enterprises that tend to make financial investment profits, and no longer make profits through the production and sales of traditional main business 23 [4]. Regarding the measurement of corporate financialization, Demir (2007) used relevant indicators, such as the proportion of financial assets held by enterprises to measure the financialization of enterprises. Domestic scholars have expanded the measurement of enterprise financialization [5]. Zhang et al. (2016) and Liu (2017) used the holding share of enterprise financial assets to measure the financialization of enterprises from a broad level and measure the profit channel of enterprises from a narrow level [6,7].

With the transformation of economic growth momentum, the relationship between enterprise financialization and enterprise innovation has attracted the attention of academic circles. At present, the academic circle has not reached an agreement on the research of whether enterprise financialization should promote or inhibit enterprise technological innovation. According to its action direction, the influence of corporate financialization on technology innovation can be divided into two studies: the promoting effect and the inhibitory effect. Scholars with a view of promotion believe that corporate financial asset allocation is based on preventive reserve motivation. By allocating financial assets to facilitate liquidity ability, enterprises can increase their financing channels so they can realize funds in a timely manner when facing external economic uncertainty. These financial assets guarantee the development of real enterprises. The appropriate financialization of enterprises can alleviate financing constraints to a certain extent. The profits obtained by enterprises from financial channels can smooth the funds needed for their production, investment, and operation, provide financial support for the technological innovation of enterprises, and help to enhance the innovation ability of enterprises and improve the profitability of entities. Bonfiglioli (2008) showed that corporate financialization enabled enterprises to obtain more investment returns, alleviate the problem of corporate financing constraints to a certain extent, create more profits for enterprises, and promote enterprise innovation investment [8]. Xu et al. (2019) discussed the impact of enterprise financialization on technological innovation from the perspectives of innovation input and innovation performance. It was found that the current financialization mainly showed a "pulling effect" on enterprise innovation. When the profitability of an enterprise was weak, the financialization of enterprises showed a "crowding out effect" on innovation investment [9]. Yang et al. (2019) found that the short-term financial investment of some idle funds of enterprises could increase the liquidity of enterprise assets, realize the preservation and appreciation of capital, and provide a financial guarantee for enterprises' investment in technological innovation and R&D. Scholars who hold the view of an inhibitory effect believe that financial investment is based on speculative profit-seeking motivation. The principal-agent theory makes enterprise ownership and management separate. Based on their own interests, the management of an enterprise invests funds in the financial sector with a high short-term yield, thus attracting capital from the entity investment [10]. Under the condition of limited resources, the financialization of enterprises affects the overall resource allocation. If enterprises use too many resources for financial assets investment, it not only shifts the business focus but also affects the innovation input. Seo et al. (2012) believed that non-financial companies investing too much of their assets in financial investment could crowd out resources for technological innovation and lead to a lack of sufficient funds for technological innovation and development [11]. Trivedi (2014) found that although the financialization of enterprises improves the financial returns of spec-

ulators, it could not improve the mismatch of financial assets, and it would also affect the efficiency of industrial investment [12]. Gleadle et al. (2014) found that the financialization of real enterprises significantly reduced investment in R&D and innovation in the current period, and the profit of enterprise financial channels inhibited technological innovation [13]. Kliman et al. (2015) analyzed changes in the financial asset structure of American listed companies and found that modern enterprises are more inclined to invest in long-term securities with weak liquidity so as to obtain higher returns [14]. Davis (2016) believed that with the transformation of the external financing structure of enterprises, the growth of financial profits and the financial profits of financial market payment could reflect the financialization tendency to some extent and lead to a decrease in corporate entity investment [15]. Cupertino et al. (2019) found that excessive financial investment made the enterprise lack enough funds to carry out product research and development innovation, thus inhibiting its technological innovation output [16]. Zhuang et al. (2022) took Chinese micro-enterprises as a research object, which showed that the main purpose of financial investment by Chinese enterprises was profit pursuit rather than precautionary savings. Fintech development aggravated the profit-seeking motivation of capital, promoted the financial investment of enterprises, and aggravated the problem of "moving from real to virtual" [17]. Xie et al. (2014) used listed company data to empirically examine the impact of manufacturing financialization on technological innovation. It was found that the excessive financialization of the manufacturing industry inhibited the ability of technological innovation. The government regulation intensified the negative impact of corporate financialization on innovation [18]. Du et al. (2017) showed that the "crowding out effect" of enterprise financialization was greater than the "reservoir effect". The income brought by financial investment did not alleviate the future underinvestment of enterprises but reduced the innovation ability of enterprises and weakened the development of the real economy [19]. Dong et al. (2021) used a sample of non-financial listed companies from 2009 to 2019 for empirical analysis. It was found that the financialization degree of enterprises had a crowding-out effect on technological innovation investment. The impact of financialization suitability on innovation investment showed a "U" dynamic transformation. If the enterprise financialization deviated from the optimal degree, it had an evidently negative impact on the enterprise innovation investment [20].

By combing the relevant literature, we found that, first of all, domestic and foreign scholars measured financialization based on different perspectives and generally measured financialization at the micro level. Secondly, domestic and foreign scholars continued to study and discuss the relationship between enterprise financialization and innovation. However, there is no consensus on the relationship between financialization and technological innovation. A number of studies have attributed the motivation of enterprise financial investment to preventive reserve motivation and speculative profit-seeking motivation. Regarding research on the influence of enterprise financialization on enterprise technological innovation, enterprise financialization could promote or inhibit enterprise technological innovation.

The main contributions of this paper are as follows: first, this paper, from the enterprise technology innovation R&D perspective, analyzes the relationship between enterprise financialization and innovation. Based on enterprise property rights and the regional financial development level, we expanded the heterogeneity analysis. This paper broadened the research fields related to enterprise financialization and innovation, helped enterprises to have a clearer understanding of the substantial impact of enterprise financialization on innovation, encouraged enterprises to actively innovate in theory and data, improved the motivation and enthusiasm of enterprise independent innovation, and provided theoretical and practical thinking for enterprises to make relevant decisions. Second, this paper adopted the intermediary effect model. By focusing on the intermediary role of financing constraints, the systematic relationship between enterprise financialization, financing constraints and technological innovation was clarified, and the internal influence transmission mechanism of financialization on technological innovation was defined. It

expanded the research scope of the impact of enterprise financialization on innovation, provided empirical evidence at the micro level for enterprise financialization on technological innovation, helped enterprises to deeply combine a financial asset allocation plan with enterprise innovation strategy, and had certain reference values for the problem of "turning from real to virtual" being faced by the government.

## 3. Theoretical Analysis and Research Hypothesis

### 3.1. Enterprise Financialization and Enterprise Technology Innovation

Enterprise investment refers to the use of funds that are held by an enterprise in order to obtain an expected return proportional to the risk within a certain period of time. Traditional finance believes that investors' investment behavior is rational; they accurately process the information obtained based on a perfect market mechanism and maximize utility as an investment goal when making investment decisions. However, the actual investment behavior of enterprises cannot be consistent with theoretical assumptions. Factors such as the asymmetry of investment market information, the adjustment of government financial policies, changes in financial market demand, an enterprise's own operating conditions, and the ability to allocate resources may all lead to the alienation of enterprise investment behavior. The financialization problem of "turning from real to virtual" in China is the result of the joint effect of the financial sector's influence on economic policy, economic growth, economic returns, and the joint effect of many micro-enterprises' financial investment behavior.

Due to the high liquidity and facilitated liquidity of financial assets, the allocation of financial assets by enterprises has the role of "reservoir" of funds, which has a positive impact on enterprise technological innovation. On the one hand, enterprise financialization can play a partially defensive role. Enterprises form part of the idle funds for short-term financial investment, increase the income of corporate financial investment, promote the liquidity of enterprise assets, and achieve the preservation and appreciation of capital. To a certain extent, financialization can prevent a shortage of funds when enterprises face the impact of an external environment so as to promote the long-term development of enterprises and make better technological innovation. The process of enterprise financialization is equivalent to a capital reservoir. The profit of financial investment improves the investment capacity of enterprise entities and increases the capital of enterprise technological innovation. On the other hand, compared with the real economy sector, a return on the investment of the financial industry is very ponderable, and the benefits brought about by financial investment are much higher than the benefits of the real economy. More experienced non-financial enterprises invest idle funds in the capital market for re-lending business. Financial investment improves the overall profitability of non-financial enterprises. Enterprises have the ability to carry out innovative activities and indirectly promote enterprise technological innovation investment.

Due to the cash flow competition between different investments, there is an alternative relationship between corporate financial investment and physical investment, and financialization has a "crowding out effect" on the technological innovation of enterprises. On the one hand, according to the principal-agent theory, the interests of the owners and managers of a company are not completely consistent in the context of the separation of ownership and management rights. Due to a high investment cost, a lack of core technology, and other problems, the profits of the real economy continue to decline, while the income of financial assets shows a continuous growth trend. The existence of the profit-seeking motive of capital makes the management of enterprises invest more capital resources in the financial field when the uncertainty of the macroeconomic environment increases. Non-financial enterprises choose to use more idle funds for financial investment, which crowds out the resources of enterprises for technological innovation, making the funds invested by enterprises in technological innovation and development obviously insufficient before inhibiting the level of technological innovation of enterprises. On the other hand, from the perspective of enterprise liquidity management, the holding of certain liquid assets by enterprises is a

crucial guarantee to maintain normal production and timely respond to external uncertain shocks. Enterprise innovation and R&D itself have the characteristics of poor liquidity, a long return on the investment period, and many uncertainties in the cycle. Innovation R&D investment is irreversible, investment risk is large, and the technological innovation R&D process also has a certain risk of failure, which intensifies the cautiousness of enterprises for the innovation investment behavior. Financial asset allocation has the characteristics of a short investment cycle and facilitates liquidity and a high return on investment. Although enterprises need to bear investment risks, non-financial enterprises tend to choose financial assets to invest in under liquidity management. There is an obvious crowding out on the relationship between technological innovation investment and financial asset investment.

To sum up, there are two views on the impact of enterprise financialization on enterprise technological innovation. Whether enterprise financialization has a positive or negative effect on enterprise technological innovation has not yet been determined. Based on the above theoretical analysis, this paper puts forward the following hypotheses:

**Hypothesis H1a:** *Corporate financialization promotes enterprise technological innovation, and there is a reservoir effect.*

**Hypothesis H1b:** *Corporate financialization inhibits enterprise technological innovation and has a crowding-out effect.*

*3.2. Heterogeneity Analysis of the Impact of Enterprise Financialization on Enterprise Technology Innovation*

Will the impact of enterprise financialization on enterprise technological innovation vary depending on the nature of enterprise property rights? Compared with the central and western regions, the level of financial development in the eastern region is high; therefore, is the impact on the financialization of entity enterprises in the eastern region on technological innovation more obvious than that of entity enterprises in the central and western regions? This paper analyzes the heterogeneity from two aspects: the nature of enterprise property rights and the financial development level of the region where the enterprise is located.

From the perspective of the nature of enterprise property rights, China's economy has long been composed of two economic sectors with different property rights, including state-owned enterprises and non-state-owned enterprises. The nature of the property rights of enterprises is different, their financial asset allocation is different, and the impact of enterprise financialization on technological innovation may be different. Specifically, state-owned enterprises have large, fixed assets and the stable development of their main business. They have institutional and policy financing advantages, which can obtain sufficient funds at a lower cost. In the case of a high return on investment in financial assets, state-owned enterprises are more inclined to make financial investments based on the profit motive of capital. State-owned enterprises have a high degree of correlation with government departments. Due to the constraints of traditional production and operation methods and the insufficient analysis of market information, state-owned enterprises usually have low technological innovation efficiency and lack continuous investment in research and development. Compared with state-owned enterprises, non-state-owned enterprises have smaller assets, and they usually face greater difficulties in transaction costs and financing constraints, from which it is difficult to obtain sufficient funds to support production through direct financing. However, the technological innovation capabilities of non-state-owned enterprises and the investment in technological innovation R&D are generally higher than those of state-owned enterprises. Therefore, the financialization of enterprises may have a greater impact on the technological innovation of non-state-owned enterprises.

From the perspective of the financial development level of the region where the enterprise is located, China's regional economic development level has long shown a pattern of being high in the east and low in the west. The financial development level of the region where the enterprise is located is different, and the impact of financialization

on technological innovation may be different. Since the reform and opening up, the total economic volume of the eastern region has maintained a leading position, and the financial industry has been actively developed. The development of the financial market has been more perfect, and the degree of information asymmetry between commercial banks, securities companies, and other financial institutions and enterprise development has been small. Therefore, non-financial enterprises in the eastern region can choose more financial investment products, and they are more inclined to invest in financial assets to obtain an income. The impact of financialization on enterprise technological innovation is more obvious than that of enterprises in the central and western regions.

Based on the above theoretical analysis, the impact of enterprise financialization on enterprise technological innovation may be heterogeneous due to the nature of enterprise property rights and the different levels of financial development in the regions where enterprises are located. This paper puts forward the following hypotheses:

**Hypothesis H2:** *The impact of enterprise financialization on enterprise technological innovation is heterogeneous in two aspects: the nature of enterprise property rights and the financial development level of the region where the enterprise is located.*

*3.3. Analysis of the Influence Mechanism of Enterprise Financialization on Enterprise Technological Innovation*

Due to the uncertainty and information asymmetry of innovation, innovation activities easily fall into external financing constraints. In the financial market led by banks, the R&D investment of Chinese enterprises faces more serious financing constraints. The influence mechanism of enterprise financialization on enterprise technology innovation this paper analyzes the influence mechanism from the perspective of an enterprise financing constraint.

There are usually two channels for enterprises to carry out technological innovation: first, enterprises adopt mergers and acquisitions to incorporate the emerging technologies of the acquirer into enterprises. Second, enterprises obtain patented technologies through independent research and development. Enterprises through the above two channels for technological innovation have a great demand for funds. Relying only on internal funds is not enough to support enterprise technological innovation; therefore, enterprises have to support technological innovation through external financing. Studies have shown that the risk of technological innovation R&D is relatively high. Compared with other investment activities of enterprises, technological innovation plays a more prominent role in the context of financing constraints.

On the one hand, when enterprises make profits from financial channels, they may partially alleviate the financing constraints of enterprises and play a role in the capital reservoir for technological innovation. This is mainly reflected in the fact that financialization alleviates information asymmetry. The main reason for the high use cost of external funds in R&D activities is that external investors have difficulty obtaining enterprise R&D information. Entity enterprises participate in external financial institutions in the form of financial investment, which has close contact with financial institutions to disclose research and development information and reduce the financing pressure on innovation activities. Second, financialization sends a good signal. Financial investment helps companies to make considerable profits in the short term and improve their return on assets. At the same time, it is also conducive for enterprises to create a good public image, obtain the affirmation of financial analysts, and enhance the confidence of external investors. Third, financialization ensures sufficient endogenous funds. The internal financing of an enterprise is superior to external financing. The financial assets invested by enterprises can be quickly realized to adjust the level of capital, ensure sufficient internal capital of enterprises, effectively improve external financing pressure, and resist the risk of innovation.

On the other hand, corporate financialization hinders the development of innovation activities by intensifying the financing pressure on enterprises. This is mainly reflected in the policy constraints. In 2017, the China Securities Regulatory Commission (CSRC) issued a Q&A on Issuance Supervision-Regulatory Requirements on Guiding and Regulating the

Financing Behavior of Listed Companies, which clearly stated that "when a non-financial listed company applied for refinancing, in principle, there shall be no financial investment such as holding a large amount of trading financial assets and financial assets with a long period of time at the end of the recent period, lending money to others, entrusted wealth management, etc.". It is stated that when enterprises make large-scale financial investments, they are subject to capital market financing constraints [21]. The second concept this is reflected in is limited to bank financing. From the perspective of bank credit financing, banks review the repayment ability and loan purpose of enterprises when conducting credit approval. Companies that invest too much money in the virtual economy rather than developing their main business make banks think there is a "false prosperity." Banks and other financial institutions reduce lending to enterprises that generate excessive financial investment, which increases the degree of financing constraints on enterprises. The deepening of financing constraints faced by enterprises reduces their technological innovation and development activities.

Judging from the above analysis, corporate financialization has had a significant impact on innovations by alleviating or exacerbating financing constraints. From this, we made the following assumptions:

**Hypothesis H3a:** *Corporate financialization strengthens the promoting effect or reduces the inhibitory effect of innovation by alleviating financing constraints.*

**Hypothesis H3b:** *Corporate financialization reduces the promoting effect or strengthens the inhibitory effect of innovation by intensifying financing constraints.*

## 4. Research Design

### 4.1. Sample Selection

In this paper, Shanghai and Shenzhen A-share listed companies from 2009 to 2020 were selected as the study samples. In order to ensure the accuracy of the research data, the following data were screened: firstly, we eliminated ST, * ST, and financial industry-listed companies so as to avoid the impact of the investment nature of financial companies on research results. Secondly, we eliminated the listed companies with missing relevant research data to avoid the impact of incomplete data on the regression results. Finally, we treat all variables with Winsorize at the 99% and 1% levels to avoid the impact of sample extremes and outliers on empirical results. After the screening, we finally retained 17,536 sample data. The investment in technological innovation and financial data of listed companies used in the empirical analysis were all from the CSMAR database.

### 4.2. Variable Selection and Description

#### 4.2.1. Dependent Variable

Starting from the financial investment behavior of enterprises, this paper discusses the impact of corporate financialization on enterprise technological innovation. According to the proposed research hypothesis, this paper measures the level of technological innovation from the perspective of technological innovation R&D investment (Rd). By referring to the research of Yang et al. (2017) and Kong et al. (2017), the total R&D investment of the enterprises was measured by a natural logarithmic value [22,23].

#### 4.2.2. Independent Variable

This paper defines enterprise financialization as the corresponding financial asset allocation behavior of enterprises. Referring to the research of Peng et al. (2018), based on the corporate balance sheet data, we measured the degree of corporate financialization by the ratio of the financial assets held by the enterprises to the total assets of the enterprises[Fin] [24]. The financial assets held by an enterprise consist of seven parts: monetary funds, trading financial assets, net investment real estate, net financial assets available for sale, net investments held to maturity, net dividends receivable, and net interest receivable.

Considering that the financialization of enterprises may have a lagging impact on enterprise technological innovation, we referred to the research of Duan et al. (2021) and used the proportion of financial assets held by enterprises (L_Fin) in the previous year to measure the degree of enterprise financialization and conduct a test of lag impact [25]. Referring to the research of Peng et al. (2022), we removed the net investment real estate and net investment from holding to maturity on the basis of the proportion of corporate finance in the original independent variable, generated a new independent variable enterprise financialization (Fin1), re-measured the proportion of corporate financial assets, and tested the robustness of alternative independent variables [2].

### 4.2.3. Control Variables

Referring to the research of Gu et al. (2018), we selected the following variables that may affect the technological innovation of enterprises as control variables. The size of the enterprise (Size) was measured by the natural logarithmic value of the total assets in the enterprise. Enterprise growth capacity (Growth) was measured by the ratio of the growth of the enterprise's operating income to the total operating income of the previous year. The Return on Total Assets (Roa) can be measured by the ratio of net profit to the share of total assets. Financial leverage (leverage) is measured by the ratio of the total liabilities in the enterprise to the owner's equity. Board structure (Board) is measured by the ratio of the number of independent directors to the number of directors. Equity concentration (Holder) is measured by the shareholding ratio of the largest shareholder [26].

### 4.2.4. Mediation Variable

Referring to the research of Xie et al. (2011), we selected an enterprise financing constraint (Fc) as the intermediary variable between enterprise financialization and enterprise technological innovation. The enterprise financing constraint was measured by the absolute value of the SA index, the larger absolute value, and a higher degree of the enterprise financing constraint [27], Table 1.

**Table 1.** Description of variables.

| Variable Category | Variable Name | Notation | Definitions and Explanations |
|---|---|---|---|
| Dependent variable | Technological innovation R&D investment | Rd | Ln (Total investment in R&D) |
| Independent variable | The proportion of corporate financial assets | Fin | Enterprises hold financial assets/total assets |
| | | L_Fin | The proportion of financial assets held by enterprises in the previous year |
| Control variables | Enterprise size | Size | Ln (total assets) |
| | Enterprise growth ability | Growth | Operating revenue growth/total revenue of last year |
| | return on total assets | Roa | Net profit/total assets |
| | financial leverage | Leverage | Total liabilities/owner's equity |
| | Board structure | Board | Number of independent directors/total number of Board of Directors |
| | Equity concentration | Holder | The largest shareholder shareholding ratio |
| Mediation variable | Enterprise financing constraints | Fc | Measured by the absolute values of the SA index SA = $-0.737 \times$ Size + $0.043 \times$ Size$^2$ − $0.04 \times$ Age |

### 4.3. Benchmark Model Construction

In order to evaluate the impact of enterprise financialization on technological innovation, a measurement model was constructed as follows:

$$Rd_{it} = \alpha_0 + \alpha_1 Fin_{it} + \alpha_k control_{it} + id_i + year_t + \varepsilon_{it} \tag{1}$$

$$Rd_{it} = \mu_0 + \mu_1 L\_Fin_{it} + \mu_k control_{it} + id_i + year_t + \varepsilon_{it} \qquad (2)$$

Model (1) takes the investment in technological innovation R&D as the dependent variable and the proportion of financial assets held by enterprises in the total assets as the independent variable. Model (2) takes the investment in technological innovation R&D as the dependent variable and the proportion of financial assets held by enterprises in the previous year as the independent variable to test the delayed impact of corporate financialization on technological innovation. $I$ represents the first enterprise in the sample. $T$ is the year. Coefficient $\alpha_1$ indicates the influence of enterprise technology innovation. If coefficient $\alpha_1$ is significantly positive, the enterprise financialization promotes enterprise technological innovation, which assumes that H1a is verified. If coefficient $\alpha_1$ is significantly negative, enterprise financialization suppresses enterprise technology innovation, which assumes H1b to be verified. $k$ represents the $k$ th control variable. The control represents all the control variables, including enterprise size, enterprise growth ability, return on total assets, financial leverage, board structure, and equity concentration. $Id_i$ represents individual fixed effect. $Year_t$ represents time fixed effect. $\varepsilon_{it}$ represents random disturbance term.

## 5. Empirical Analysis

### 5.1. Descriptive Statistical Analysis

Table 2 shows the descriptive statistics of the main variables. According to Table 2, the dependent variable enterprise technology innovation R&D average was very close to the median, with a minimum of 11.603 and a maximum of 22.357. These data show that there are different degrees of difference in the technological innovation R&D investment level of China's non-financial listed companies, and the degree of importance to technological innovation R&D investment varied greatly among enterprises. The average variable enterprise financialization was 0.041, which shows the enterprises holding financial assets of total assets of 4.1%. It had a minimum of 0 and a maximum of 0.520, which shows that China's non-financial listed companies' financial investment occupies an important position. The difference in financial asset allocation was obvious. Some companies tended to have a high position allocation of financial assets, and enterprise financialization and enterprise technology innovation-related issues were very necessary. For the relevant control variables, the difference between the mean value and the median of the most controlled variables was small, which indicates that the relevant control variables were valued within a reasonable range.

**Table 2.** Descriptive statistics for the main variables.

| Variables | Obs | Mean | Median | SD | Min | Max |
|---|---|---|---|---|---|---|
| Rd | 17536 | 17.830 | 17.844 | 1.559 | 11.603 | 22.357 |
| Fin | 17536 | 0.041 | 0.012 | 0.072 | 0.000 | 0.520 |
| L_Fin | 12738 | 0.037 | 0.011 | 0.065 | 0.000 | 0.488 |
| Size | 17536 | 6.127 | 5.925 | 1.297 | 3.745 | 10.484 |
| Growth | 17536 | −0.330 | −0.106 | 7.264 | −83.827 | 41.532 |
| Roa | 17536 | 0.051 | 0.046 | 0.050 | −0.329 | 0.239 |
| Leverage | 17536 | 0.900 | 0.610 | 0.908 | 0.021 | 6.751 |
| Board | 17536 | 0.375 | 0.333 | 0.054 | 0.250 | 0.571 |
| Holder | 17536 | 34.534 | 32.590 | 14.792 | 8.010 | 75.250 |

Table 3 shows the results of Pearson's correlation test for the main variable. As we can see from Table 3, the negative relationship between enterprise financialization and enterprise technology innovation variables is negative, which initially supports the H1b hypothesis proposed above. The correlation coefficients between most variables are significant, and the absolute value of the correlation coefficient between variables is basically less than 0.5, indicating that there was no serious collinearity among the variables studied in this paper.

**Table 3.** Association tests for the primary variables.

|  | Rd | Fin | Size | Growth | Roa | Leverage | Board | Holder |
|---|---|---|---|---|---|---|---|---|
| Rd | 1 |  |  |  |  |  |  |  |
| Fin | −0.069 *** | 1 |  |  |  |  |  |  |
| Size | 0.513 *** | −0.063 *** | 1 |  |  |  |  |  |
| Growth | −0.00600 | 0.00600 | 0.00500 | 1 |  |  |  |  |
| Roa | 0.062 *** | 0.068 *** | −0.097 *** | 0.402 *** | 1 |  |  |  |
| Leverage | 0.138 *** | −0.125 *** | 0.536 *** | −0.055 *** | −0.328 *** | 1 |  |  |
| Board | 0.037 *** | 0.039 *** | 0.025 *** | 0.00100 | −0.00200 | 0.017 *** | 1 |  |
| Holder | 0.026 *** | −0.038 *** | 0.190 *** | 0.053 *** | 0.108 *** | 0.095 *** | 0.066 *** | 1 |

Note: The numbers in the table are the correlation coefficient between the relevant variables, *** is significant at 1% levels, respectively.

### 5.2. Benchmark Regression Results

Table 4 shows the benchmark regression results of the impact of corporate financialization on corporate technological innovation. Column (1) shows the estimated results of the impact of corporate financialization on R&D investment in technological innovation in the current period. After controlling for the individual effect and time effect, the regression coefficient of the independent variable *Fin* was −0.521, which was significantly negative at the 1% level. The data indicate that corporate financialization has an obvious inhibitory effect on technological innovation R&D investment. The financial investment of enterprises occupies the resources of technological innovation investment, and the higher the degree of corporate financialization, the less the enterprise invested in technological innovation research and development. This regression result verifies hypothesis H1b proposed above. Column (2) shows the regression results of the influence of corporate financialization on the one-stage lag of R&D investment in technological innovation. The regression coefficient of the independent variable *L_Fin* was −0.461, which was also significantly negative at the 1% level. The data indicate how corporate financialization had a lagging inhibitory effect on corporate technological innovation R&D investment.

**Table 4.** Benchmark regression results.

| Variables | (1)<br>Rd | (2)<br>Rd |
|---|---|---|
| Fin | −0.521 *** |  |
|  | (−4.20) |  |
| L_Fin |  | −0.461 *** |
|  |  | (−3.41) |
| Size | 0.716 *** | 0.735 *** |
|  | (23.61) | (21.68) |
| Leverage | −0.092 *** | −0.091 *** |
|  | (−3.81) | (−3.19) |
| Growth | −0.005 *** | −0.004 *** |
|  | (−5.07) | (−4.06) |
| Roa | 1.108 *** | 0.743 *** |
|  | (5.62) | (3.69) |
| Holder | 0.002 | 0.002 |
|  | (1.10) | (0.80) |
| Board | −0.468 ** | −0.325 * |
|  | (−2.44) | (−1.67) |
| Observations | 16995 | 12266 |
| $R^2$ | 0.894 | 0.918 |
| id FE | Yes | Yes |
| year FE | Yes | Yes |

Note: Robust t-statistics in parentheses *** $p < 0.01$, ** $p < 0.05$, * $p < 0.1$.

### 5.3. Heterogeneity Analysis

Based on the heterogeneity theoretical analysis of the influence of enterprise financialization on enterprise technological innovation, the research samples were divided into the following groups: first, according to the property rights of enterprises, the research samples

were divided into state-owned enterprises and non-state-owned enterprises. Second, according to the financial development level of the region where the enterprises were located, the research samples were divided into enterprises in the western region, enterprises in the central region, and enterprises in the eastern region. We used the above subsamples separately for benchmark regression.

Table 5 shows the results of subsample regression according to the property rights of enterprises. It can be seen from Table 5 that the financialization of both state-owned enterprises and non-state-owned enterprises had a crowding-out effect on technological innovation's R&D input, but there were certain differences in the significance of negative effects. Column (1) is the regression result of the influence of financialization on the state-owned enterprises of a technological innovation R&D input. Column (2) is the regression result of the influence of financialization on non-state-owned enterprises and technological innovation R&D input. As can be seen from the estimated coefficient of the independent variable *Fin*, state-owned enterprises and non-state-owned enterprises were more motivated by capital profit-seeking financial asset allocation. Since non-state-owned enterprises have stronger technological innovation abilities and a higher innovation efficiency than state-owned enterprises, the negative effect of enterprise financialization on enterprise technological innovation input was significant at the 1% level.

**Table 5.** Subsample regression according to the nature of enterprise property rights.

| Variables | (1)<br>Rd | (2)<br>Rd |
|---|---|---|
| Fin | −0.461 | −0.385 *** |
| | (−1.09) | (−3.22) |
| Size | 0.810 *** | 0.721 *** |
| | (10.35) | (22.81) |
| Leverage | −0.028 | −0.112 *** |
| | (−0.70) | (−3.52) |
| Growth | −0.003 | −0.006 *** |
| | (−1.38) | (−5.65) |
| Roa | 1.601 *** | 1.056 *** |
| | (2.88) | (5.18) |
| Holder | −0.002 | 0.003 |
| | (−0.42) | (1.12) |
| Board | 0.048 | −0.788 *** |
| | (0.14) | (−3.75) |
| Observations | 5248 | 11667 |
| $R^2$ | 0.896 | 0.900 |
| id FE | Yes | Yes |
| year FE | Yes | Yes |

Note: brackets t statistics, *** $p < 0.01$.

Table 6 shows the results of sub-sample regression according to the financial development level of the region where the enterprises are located. As can be seen from Table 6, with different financial development levels in the regions where enterprises were located, the crowding out effect of financialization on technological innovation R&D investment of enterprises was different. Column (1) shows the regression results of the influence of enterprise financialization on enterprise technological innovation R&D investment in western China. Column (2) shows the regression results of the influence of enterprise financialization on enterprise technological innovation R&D investment in central China. Column (3) shows the regression result of the influence of enterprise financialization on enterprise technological innovation R&D investment in eastern China. Due to the relatively low level of financial development in central and western regions, the negative effect of corporate financialization on technological innovation R&D investment was relatively small. The financial development level of the eastern region was relatively high. The eastern region has financing convenience and rich types of financial products, and enterprises tend to use

part of the capital to invest in financial assets so as to obtain higher investment returns. Therefore, financial enterprises in the eastern region have a significant negative impact on investment in technological innovation, research, and development.

**Table 6.** Regression of samples according to the financial development level of the region where the enterprise is located.

|  | (1) | (2) | (3) |
|---|---|---|---|
| **Variables** | **Rd** | **Rd** | **Rd** |
| Fin | −0.246 | −0.427 | −0.499 *** |
|  | (−0.44) | (−1.28) | (−3.68) |
| Size | 0.663 *** | 0.838 *** | 0.706 *** |
|  | (7.21) | (10.23) | (20.90) |
| Leverage | −0.144 ** | −0.075 | −0.085 *** |
|  | (−2.08) | (−1.15) | (−3.01) |
| Growth | −0.003 | −0.007 *** | −0.004 *** |
|  | (−1.13) | (−3.25) | (−3.83) |
| Roa | 1.396 * | 1.220 ** | 1.010 *** |
|  | (1.73) | (2.27) | (4.76) |
| Holder | −0.001 | 0.008 | 0.001 |
|  | (−0.13) | (1.32) | (0.42) |
| Board | −0.113 | −0.661 | −0.497 ** |
|  | (−0.19) | (−1.31) | (−2.27) |
| Observations | 1942 | 2484 | 12515 |
| $R^2$ | 0.878 | 0.886 | 0.900 |
| id FE | Yes | Yes | Yes |
| year FE | Yes | Yes | Yes |

Note: brackets t statistic, *** $p < 0.01$, ** $p < 0.05$, * $p < 0.1$.

To sum up, the influence of enterprise financialization on enterprise technology innovation has heterogeneity in both the nature of enterprise property rights and the financial development level in the region where the enterprise is located, which verifies hypothesis H2.

*5.4. Impact Mechanism Analysis*

5.4.1. Construction of the Mediation Effect Model

Benchmark regression analysis verified that there was a negative correlation between firm financialization and firm technological innovation. If so, how does firm financialization affect technological innovation? Based on the analysis of the influence mechanism of corporate financialization on corporate technological innovation mentioned above, this paper chose the enterprise financing constraint as the intermediary variable between enterprise financialization and enterprise technological innovation. The following mediation effect model was constructed by referring to the step-by-step method proposed by Baron and Kenny (1986) to test the mediation effect [28]:

$$Rd_{it} = a_0 + a_1 Fin_{it} + a_k control_{it} + id_i + year_t + \varepsilon_{it} \tag{3}$$

$$Fc_{it} = b_0 + b_1 Fin_{it} + b_k control_{it} + id_i + year_t + \varepsilon_{it} \tag{4}$$

$$Rd_{it} = c_0 + c_1 Fin_{it} + c_2 Fc_{it} + c_k control_{it} + id_i + year_t + \varepsilon_{it} \tag{5}$$

$Rd_{it}$ is the technological innovation R&D input of the dependent variable. $Fin_{it}$ is the proportion of the financial assets of the dependent variable. $Fc_{it}$ is the financing constraint of the intermediary variable. By referring to the mediation effect test process proposed by Wen et al. (2014), the regression coefficient $a_1$ of Fin in Equation (3) was observed to test whether enterprise financialization had a significant impact on enterprise technological innovation. We observed the regression coefficient $b_1$ of Fin in Equation (4) to test whether corporate financialization had a significant impact on financing constraints. We observed the regression coefficient $c_1$ of Fin in Equation (5) and tested whether corporate finan-

cialization and financing constraints had a significant impact on corporate technological innovation [29]. When coefficient $a_1$ was significant, and if coefficient $b_1$ and coefficient $c_2$ were both significant, this indicated that corporate financing constraints were playing an intermediary role between corporate financialization and corporate technological innovation. In this case, when coefficient $c_1$ was not significant, it indicated that corporate financing constraints had a complete intermediary effect between corporate financialization and corporate technological innovation. When the coefficient $c_1$ was significant, if the symbols of $b_1{}^*c_2$ and $c_1$ were the same, then corporate financing constraints had a partial mediating effect between corporate financialization and corporate technological innovation. If the symbols of $b_1{}^*c_2$ and $c_1$ were different, then corporate financing constraints had a masking effect between corporate financialization and corporate technological innovation.

### 5.4.2. Estimation Results of the Mediation Effect Model

Table 7 shows the regression results of the mediation effect model. Column (2) is listed as the regression result of the independent variable *Fin* on the intermediate variable *Fc*. Coefficient $b_1$ was significantly positive at the 1% level, which indicates that financial investment intensifies corporate financing constraints, making it more difficult for enterprises to achieve external financing. Column (3) is listed as the regression result of the dependent variable *Rd* on the independent variable Fin and intermediate variable *Fc*. When both coefficients $c_1$ and $c_2$ were significantly negative, it indicated that the intensification of corporate financing constraints significantly inhibited corporate technological innovation. When the symbols of $b_1{}^*c_2$ and $c_1$ were the same, it indicated that corporate financing constraints had an intermediary effect between corporate financialization and corporate technological innovation. By comparing the regression coefficient of the independent variable Fin in column (1) and column (3), it could be seen that after the addition of the intermediate variable *Fc*, the inhibitory effect of corporate financialization on R&D investment in corporate technological innovation decreased, indicating that corporate financing constraints bore part of the intermediary effect between corporate financialization and corporate technological innovation, and corporate financialization behavior intensified financing constraints to a certain extent and inhibited the technological innovation of enterprises. The above analysis verifies hypothesis H3b.

**Table 7.** Results of the mediation affect model regression.

| Variables | (1) Rd | (2) Fc | (3) Rd |
|---|---|---|---|
| Fin | −0.521 *** | 0.064 *** | −0.467 *** |
| | (−4.20) | (5.63) | (−3.76) |
| Fc | | | −0.849 *** |
| | | | (−3.47) |
| Size | 0.716 *** | 0.206 *** | 0.891 *** |
| | (23.61) | (49.19) | (13.89) |
| Leverage | −0.092 *** | 0.003 | −0.090 *** |
| | (−3.81) | (1.21) | (−3.74) |
| Growth | −0.005 *** | −0.000 ** | −0.005 *** |
| | (−5.07) | (−2.06) | (−5.18) |
| Roa | 1.108 *** | 0.004 | 1.112 *** |
| | (5.62) | (0.27) | (5.62) |
| Holder | 0.002 | −0.001 *** | 0.001 |
| | (1.10) | (−4.02) | (0.73) |
| Board | −0.468 ** | −0.000 | −0.468 ** |
| | (−2.44) | (−0.01) | (−2.47) |
| Observations | 16995 | 16995 | 16995 |
| $R^2$ | 0.894 | 0.987 | 0.895 |
| id FE | Yes | Yes | Yes |
| year FE | Yes | Yes | Yes |

Note: brackets t statistic, *** $p < 0.01$, ** $p < 0.05$.

*5.5. Robustness Test*

5.5.1. Replace the Independent Variables

In order to more comprehensively measure the impact of firm financialization on firm technological innovation, in the robustness test part, the independent variables were replaced first. On the one hand, considering the particularity of the real estate industry in China's economic development, the domestic companies listed in the financial assets investment cycle were usually shorter than the real estate investment cycle; real estate investment was recognized by most Chinese investors' capital value, so the entity enterprise asset allocation in real estate investment might not be entirely for speculative motives. On the other hand, the net investment held by enterprises to maturity usually had a long-term nature. The long-term financial investment made by enterprises could be based on their long-term development strategy, which does not belong to the short-term speculative behavior of enterprises. In the robustness test, the net investment real estate and net hold-to-maturity investment were removed from the calculation formula of the original independent variable corporate financial assets ratio (Fin) to generate a new independent variable corporate financial assets ratio (Fin1). This was measured by the ratio of the sum of monetary funds, trading financial assets, net financial assets available for sale, net dividends receivable, and net interest receivable to the total assets of the enterprise. Table 8 shows the regression result for replacing independent variables. After replacing independent variables, the regression coefficient of independent variable Fin1 was −0.394, which was significantly negative at the 1% level. The data indicate that enterprise financialization still has a significant negative impact on enterprise technological innovation R&D investment and that the negative effect also has a lag. This test remained consistent with the original conclusion. The robustness of the original conclusion is proved.

**Table 8.** Results of regression of replacement independent variables.

| Variables | (1)<br>Rd | (2)<br>Rd |
|---|---|---|
| Fin1 | −0.394 *** | |
| | (−3.26) | |
| L_Fin1 | | −0.324 ** |
| | | (−2.34) |
| Size | 0.719 *** | 0.738 *** |
| | (23.65) | (21.69) |
| Leverage | −0.093 *** | −0.092 *** |
| | (−3.84) | (−3.22) |
| Growth | −0.005 *** | −0.004 *** |
| | (−5.12) | (−4.10) |
| Roa | 1.121 *** | 0.748 *** |
| | (5.69) | (3.71) |
| Holder | 0.002 | 0.002 |
| | (1.12) | (0.83) |
| Board | −0.469 ** | −0.325 * |
| | (−2.44) | (−1.67) |
| Observations | 16995 | 12266 |
| $R^2$ | 0.894 | 0.917 |
| id FE | Yes | Yes |
| year FE | Yes | Yes |

Note: brackets t statistic, *** $p < 0.01$, ** $p < 0.05$, * $p < 0.1$.

5.5.2. Sample Subinterval Estimation

Considering the US subprime crisis in 2008 and the subsequent "four trillion investment" policy, this may have had a sustained impact on the financial investment and technological innovation of Chinese non-financial enterprises. In order to exclude the impact of special events on the research conclusions, sample data from two years after the subprime crisis were excluded from the robustness test. In other words, we excluded

the sample data from 2009 and 2010 for sample sub-interval estimation. Table 9 shows the regression results of the sample subinterval. After removing the sample data in 2009 and 2010, corporate financialization still had a significant negative impact on corporate technological innovation R&D investment, and the negative effect still had a lag. This test remains consistent with the original conclusion. The robustness of the original conclusion was proved.

**Table 9.** Results of sample subinterval regression.

| Variables | (1) Rd | (2) Rd |
|---|---|---|
| Fin | −0.483 *** | |
| | (−4.08) | |
| L_Fin | | −0.457 *** |
| | | (−3.51) |
| Size | 0.744 *** | 0.741 *** |
| | (25.08) | (21.58) |
| Leverage | −0.106 *** | −0.087 *** |
| | (−4.19) | (−2.92) |
| Growth | −0.004 *** | −0.004 *** |
| | (−4.97) | (−3.91) |
| Roa | 0.947 *** | 0.693 *** |
| | (5.12) | (3.55) |
| Holder | 0.001 | 0.002 |
| | (0.73) | (0.85) |
| Board | −0.334 * | −0.278 |
| | (−1.80) | (−1.44) |
| Observations | 15951 | 11974 |
| $R^2$ | 0.911 | 0.922 |
| id FE | Yes | Yes |
| year FE | Yes | Yes |

Note: brackets t statistic, *** $p < 0.01$, * $p < 0.1$.

### 5.5.3. Instrumental Variable

Since there may be endogenous problems caused by the reverse causality between the variables of corporate financial investment and technological innovation R&D level. In order to better mitigate the impact of endogenous problems, an instrumental variable method (two-stage least square method) was used in this paper to reduce the impact of endogenous problems. Referring to the research of Wang et al. (2017), we considered the investment income for the enterprise foreign investment income, including the enterprise during a certain period of the accounting foreign investment dividend income, bond interest income, and those associated with other units of profits. Its main enterprise's internal financial asset allocation level and enterprise technology innovation research and development activities would not have a direct impact on the enterprise's technological innovation and R&D input [30]. It can satisfy the correlation and exogeneity hypothesis of instrumental variables well. In this paper, the ratio of investment income to operating income (Inv) was selected as an instrumental variable to solve the endogenous bias caused by reverse causation, and the endogeneity test was conducted by using the two-stage least square method.

Table 10 shows the estimation results after regression using the two-stage least square method. The regression results of the first stage show that the regression coefficient of the instrumental variable Inv and independent variable Fin was 0.405 and had a positive significance at the 1% level. The test value of the Kleibergen-Paap rk Wald F statistic was 347.946. If the value was much larger than 10, it indicated that the tool variable Inv was recognizable and not weak. The regression results of the second stage showed that the regression coefficient of the independent variable Fin and dependent variable Rd was −14.856, which was still significantly negatively correlated at the level of 1%. This test is

consistent with the original regression results, which could more effectively weaken the effect of endogeneity.

**Table 10.** Results of regression with the instrumental variable method.

| Variables | (1)<br>First Stage<br>Fin | (2)<br>Second Stage<br>Rd |
|---|---|---|
| Inv | 0.405 *** | |
| | (0.013) | |
| Fin | | −14.856 *** |
| | | (0.741) |
| Size | −0.001 *** | 0.757 *** |
| | (0.000) | (0.011) |
| Growth | −0.000 ** | −0.014 *** |
| | (0.000) | (0.002) |
| Roa | 0.048 *** | 4.065 *** |
| | (0.012) | (0.289) |
| Leverage | −0.007 *** | −0.407 *** |
| | (0.001) | (0.018) |
| Board | 0.056 *** | 1.739 *** |
| | (0.010) | (0.233) |
| Holder | −0.000 ** | −0.012 *** |
| | (0.000) | (0.001) |
| Constant | 0.030 *** | 13.713 *** |
| | (0.005) | (0.109) |
| Kleibergen-Paap rk Wald F | | 347.946 |
| Observations | 17536 | 17536 |

Note: brackets t statistic, *** $p < 0.01$, ** $p < 0.05$.

## 6. Conclusions

This paper empirically examines the impact of corporate financialization on corporate technological innovation based on the panel data of listed A-share enterprises in Shanghai and Shenzhen from 2009 to 2020. First of all, this paper tested the influence of enterprise financialization on technological innovation with a two-way fixed effect model. Secondly, the paper analyzed heterogeneity from the nature of the enterprise's property rights and the level of financial development in the region where the enterprise was located. Finally, the paper tested the mediation effect of financing constraints through the three-step method of the mediation effect. The research results were as follows: first, enterprise financialization has a significant crowding out effect on investment in enterprise technological innovation. The larger scale of financial assets allocated by enterprises, the more serious crowding-out effect on enterprise R&D innovation, and the crowding out effect has lag. Second, the heterogeneity analysis showed that compared with state-owned enterprises, the financialization of non-state-owned enterprises had a greater crowding effect on enterprise technological innovation. Compared with the central and western regions, the level of financial development was higher, and the negative effect of enterprise financialization on enterprise technological innovation was greater. Third, the analysis of the influence mechanism further showed that there is some intermediary effect between financing constraint and enterprise financialization and enterprise technology innovation. The excessive allocation of financial assets increases the external financing constraints of enterprises and, thus, inhibits technological innovation.

Combining the findings of this paper, policy recommendations can be put forward at both government and enterprise levels. From the government level, first of all, the government should deepen the reform of the financial industry system to serve real economic development. The gradual overcapacity of the real economy and the high profits of the financial sector are important reasons for the influx of Chinese real enterprises into the financial sector. The development of the financial industry should serve the development of

the real economy rather than squeezing out real investment. On the one hand, the government should further promote supply-side structural reform, optimize the environment for financial development, promote the combination of an effective market and the competent government, and give full play to the role of the financial market as a "reservoir". On the other hand, the government should encourage the development of the real economy, promote the transformation of the real industry into a dynamic force, build innovation platforms for enterprises, help them carry out technological innovation activities, actively promote the establishment of a modern industrial system with scientific and technological innovation as its core competitiveness, deepen the reform of the institutional mechanism for the protection of intellectual property rights, and stimulate the vitality of the main body of market innovation. Secondly, the government should improve the efficiency of financial services and improve the financing difficulties of enterprises. According to the conclusion, the deepening of the financialization of enterprises could aggravate financing constraints and seriously restrict the innovation and development of enterprises. Therefore, on the one hand, the government should optimize the financing structure, increase the financing channels of real enterprises, and solve the financing difficulties of small and medium-sized enterprises. On the other hand, the government should increase policy support for real enterprises, adopt policies such as industrial support or tax incentives to alleviate the financing difficulties faced by enterprise innovations, ensure resource investment in enterprise innovation activities, and achieve high-quality economic development. Finally, the government should strengthen the supervision of financial market investment and build a financial monitoring mechanism for enterprises. The government should strictly control the scale of the financial asset allocation of enterprises, curb the unlimited expansion of capital, pay attention to prevent financial risks, build a multi-tiered financial regulatory system, identify the problem of "moving from real to virtual" in the process of economic operations, and create a good business environment for real enterprises.

At the enterprise level, enterprises should establish the correct sense of management and formulate a long-term sustainable development strategy. Enterprises should base themselves on the development of their main business, make reasonable non-productive investments according to their own development needs, and avoid the impulse of financial investment. Entity enterprises should pay attention to capital innovation, technology innovation research and development achievements as an important indicator of management performance appraisal, where the incentive management of more enterprise asset allocations in technology innovation research, and development investment can reduce manage excessive financial investment, avoid excessive financialization problems, guaranteeing the advancement of technological innovation and achieving high-quality development. Secondly, enterprises should establish and improve the internal risk management system and cope with financial investment and the technological innovation between each link to establish a comprehensive risk identification and management system. This can better predict, evaluate and control the risk of enterprise financial investment and technological innovation, minimize the risks faced by enterprises, and enhance the foundation of enterprise technological innovation.

**Author Contributions:** Conceptualization, T.Z. and X.S.; methodology, T.Z.; software, X.S.; validation, T.Z. and X.S.; formal analysis, X.S.; investigation, T.Z.; resources, T.Z.; data curation, X.S.; writing—original draft preparation, X.S.; writing—review and editing, T.Z. All authors have read and agreed to the published version of the manuscript.

**Funding:** This research received no external funding.

**Institutional Review Board Statement:** Not applicable.

**Informed Consent Statement:** Not applicable.

**Data Availability Statement:** Not applicable.

**Conflicts of Interest:** The authors declare no conflict of interest.

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
