# Peer review of "Enterprise Financialization and Technological Innovation: An Empirical Study Based on A-Share Listed Companies Quoted on Shanghai and Shenzhen Stock Exchange"

_fintech, doi:10.3390/fintech2020016_

Round 1

Reviewer 1 Report

The paper presents an interesting approach to an important problem "financialization and technological innovation". Overall, this was an interesting and valuable paper that was a pleasure to read. However, I have minor comments that can be easily resolved. First, I think that most of the references are not recent. Authors can add more recent papers in the field of financialization and technological innovation and enrich their literature review.  Second, as here is a problem of endogeneity and authors used instrumental method, it would be better to employ the GMM system estimates and make the specification tests of endogeneity (Wu-Hausman test, J-statistic and Sargan test). Finally, the conclusion needs to be extended to include more implications for policymakers. Unfortunately, this paper needs minor corrections to be suitable for publication in the FinTech Journal.

Author Response

Dear reviewer:

Thank you for your decision and constructive comments on my manuscript entitled "Enterprise financialization and technological innovation: An Empirical Study Based on A-share Listed Companies Quoted on Shanghai and Shenzhen Stock Exchange". (ID: fintech-2260259). We have carefully considered the suggestion of reviewer and made some changes. 

We have tried our best to improve and make some changes in the manuscript. These changes will not influence the content and framework of the paper. The red part that has been revised according to your comments. Revision notes, point-to-point, are given as follows:

Reviewer1: Response to comment: most of the references in the paper are not up to date, and the author can add more recent papers in the field of financialization and technological innovation. Thank you for the references suggested. We add new references in the literature review section and improv the literature review content.

Reviewer 2: Response to comment: Since this is an endogeneity problem and the authors use an instrumental approach, it is best to use the GMM system for estimating and performing normative tests for endogeneity (Wu-Hausman test, J-statistic and Sargan test). Thanks for the endogeneity question. Considering that the model set in this paper is a two-way fixed effect model, there is no over-identification problem, we report the results of 2SLS regression in the results. After receiving the revision comments, we conducted a GMM check on the system, and we are sorry that we did not get the expected results. In the following academic research, we will continue to try to discuss the problem of endogeneity.

Reviewer 3: Response to comment: The conclusions need to be expanded to include more implications for policymakers. This paper expands the research conclusions and supplements the impact of the research conclusions on the government and enterprises. Additionally, we have improved decision suggestions.

Thank you so much for your attention and time. Looking forward to hearing from you.

Your sincerity,

Corresponding Author:

Name: Sun Xinyu

e-mail:104753210111@henu.edu.cn

Reviewer 2 Report

Dear Authors,

you have taken up an interesting topic; nevertheless, it needs refinement.

Here are my comments:

1. you need to refine the template of the article according to the guidelines of the publisher - at the moment you have not used the correct template 

2. at the introduction is 0 

3. on the 2nd page the first footnote is No. 25 - also the chronology of the literature has not been applied

4. you should refine the purpose, which should be clearly visible in the abstract and transferred in the development to the introduction

5. please use the correct numbering of subsections throughout the article

6. please expand on the contribution to science and describe why the issue you are addressing is important 

7. the discussion section is missing, indicating what other authors have said on the subject and what your article contributes to science on this issue

The article needs to be completed and corrected in order to be published

Author Response

Dear reviewer:

Thank you for your decision and constructive comments on my manuscript entitled "Enterprise financialization and technological innovation: An Empirical Study Based on A-share Listed Companies Quoted on Shanghai and Shenzhen Stock Exchange". (ID: fintech-2260259). We have carefully considered the suggestion of reviewer and made some changes.

We have tried our best to improve and make some changes in the manuscript. These changes will not influence the content and framework of the paper. The red part that has been revised according to your comments. Revision notes, point-to-point, are given as follows:

Reviewer1: Response to comment: you need to refine the template of the article according to the guidelines of the publisher - at the moment you have not used the correct template. Thank you for the template of the article suggested. We improved the template of the article by referring to the journal requirements and the articles officially published in this journal.

Reviewer2: Response to comment: at the introduction is 0. We have revised the introduction.

Reviewer3: Response to comment: on the 2nd page the first footnote is No. 25 - also the chronology of the literature has not been applied. We are very sorry for our negligence of literature citation problem. We have revised the way references are used.

Reviewer4: Response to comment: you should refine the purpose, which should be clearly visible in the abstract and transferred in the development to the introduction. In the introduction part, we explained the research purpose of this paper and annotated it in red font.

Reviewer5: Response to comment: please use the correct numbering of subsections throughout the article. We have revised the correct numbering of subsections throughout the article.

Reviewer6: Response to comment: please expand on the contribution to science and describe why the issue you are addressing is important. In the literature review section, we described the research contributions of this paper and annotated them in red.

Reviewer7: Response to comment: the discussion section is missing, indicating what other authors have said on the subject and what your article contributes to science on this issue. We added the literature review, further summarized the research results of existing scholars, and elaborated the research contribution of this paper.

Thank you so much for your attention and time. Looking forward to hearing from you.

Your sincerity,

Corresponding Author:

Name: Sun Xinyu

e-mail:sxy17839973472@163.com

Reviewer 3 Report

Indeed the authors improved significantly the quality of the manuscript which now contains all the information required to gauge by the reader the soundness of the analysis and appraisal the relevance of results. As for both of them , it is still my convincement that this paper deserves publication. Nevertheless, there are still some minor and major shortcomings that the authors need to address.

I will list them below:

1)Authors through all the text refer to "energy consumption" while addressing share of RES over electricity consumption. Thus: it is not energy, but electricity, and it is not ALL power generation, but only RES.

2) All regressors are not normalized, except RES. I suggest, for consistency, at least to test the electricity (total) and RES not as a share, but as a value

3) It is still missing a motivation why fin-tech and IT should impact on emissions (and relative literature)

4) With respect to the state of art: authors claim to be the first to purportedly provide a study on the relationship  between fin-tech and eissions/sustainability, but i doubt. Below some examples I found after  a quick and dirty search. Thus, I suggest the authors to develop a literature review on the topic (now entirely missing):

Tao, R., Su, C. W., Naqvi, B., & Rizvi, S. K. A. (2022). Can Fintech development pave the way for a transition towards low-carbon economy: A global perspective. Technological Forecasting and Social Change, 174, 121278. Muhammad, S., Pan, Y., Magazzino, C., Luo, Y., & Waqas, M. (2022). The fourth industrial revolution and environmental efficiency: The role of fintech industry. Journal of Cleaner Production, 381, 135196. Croutzet, A., & Dabbous, A. (2021). Do FinTech trigger renewable energy use? Evidence from OECD countries. Renewable Energy, 179, 1608-1617. Deng, X., Huang, Z., & Cheng, X. (2019). FinTech and sustainable development: Evidence from China based on P2P data. Sustainability, 11(22), 6434.

Author Response

Dear reviewer:

Thank you for your decision and constructive comments on my manuscript entitled "Enterprise financialization and technological innovation: An Empirical Study Based on A-share Listed Companies Quoted on Shanghai and Shenzhen Stock Exchange". (ID: fintech-2260259). We have carefully considered the suggestion of reviewer and made some changes.

We have tried our best to improve and make some changes in the manuscript. These changes will not influence the content and framework of the paper. The red part that has been revised according to your comments. According to the opinions of reviewers, we have added the research literature related to enterprise financialization and technological innovation. The purpose, contribution and conclusion of this paper are supplemented.

Thank you so much for your attention and time. Looking forward to hearing from you.

Your sincerity,

Corresponding Author:

Name: Sun Xinyu

e-mail:sxy17839973472@163.com

Reviewer 4 Report

The article is interesting and correctly introduces the research gap. However, it requires correction and substantive supplementation.

1. The purpose of the article should be clearly defined and start with the word "the purpose of the article is...". This approach will allow the potential reader to find out what the authors are trying to achieve. Currently, the definition of the problem through questions is insufficient.

2.Unfortunately, I suggest that you expand your selection of literature. The current one is quite narrow and insufficient.

3. Three items are missing in the "conclusion" part. (1) Authors should make a broad reference to what their research contributes to the literature review presented and refer to how the literature review extends their research (what distinguishes their research).

Author Response

Dear reviewer:

Thank you for your decision and constructive comments on my manuscript entitled "Enterprise financialization and technological innovation: An Empirical Study Based on A-share Listed Companies Quoted on Shanghai and Shenzhen Stock Exchange". (ID: fintech-2260259). We have carefully considered the suggestion of reviewer and made some changes.

We have tried our best to improve and make some changes in the manuscript. These changes will not influence the content and framework of the paper. The red part that has been revised according to your comments. Revision notes, point-to-point, are given as follows:

Reviewer1: Response to comment: the purpose of the article should be clearly defined and start with the word "the purpose of the article is...". This approach will allow the potential reader to find out what the authors are trying to achieve. Currently, the definition of the problem through questions is insufficient. We made a supplement to the introduction part of the paper, in which we stated the research purpose of this paper.

Reviewer2: Response to comment: unfortunately, I suggest that you expand your selection of literature. The current one is quite narrow and insufficient. We have made improvements to the paper, including additions to the introduction and literature review, additions to the analysis of influence mechanisms, and expansions to the conclusions and policy recommendations.

Reviewer3: Response to comment: three items are missing in the "conclusion" part. (1) Authors should make a broad reference to what their research contributes to the literature review presented and refer to how the literature review extends their research (what distinguishes their research). We have improved the literature review, summarized the research achievements of the existing scholars, and elaborated the research contribution of this paper.

Thank you very much for your attention and time. Looking forward to hearing from you.

Yours sincerely,

Name: Xinyu Sun

Email: sxy17839973472@163.com

Round 2

Reviewer 2 Report

Dear Authors,

Thank you for getting back to me. Nevertheless, it would be best if you still expanded the literature. I suggest you expand your literature selection, as the current one is relatively narrow and insufficient.

Author Response

Dear editor,

Thank you for your decision-making and constructive comments on my manuscript entitled "Enterprise financialization and technological innovation: An Empirical Study Based on A-share Listed Companies Quoted on Shanghai and Shenzhen Stock Exchange". (ID: fintech-2260259). We have carefully considered the reviewers' suggestions and made some changes.

Every effort has been made to improve and make changes to the manuscript. These changes do not affect the content and framework of the paper. Yellow section modified according to your comments. We add to the research literature related to firm financialization and technological innovation.

Thank you so much for your attention and time. Looking forward to hearing from you.

your sincerity,

Corresponding Author:

Name: Sun Xinyu

Email: sxy17839973472@163.com

Reviewer 4 Report

I accept the correction of the article.

Author Response

Dear editor:

Thank you for your decision-making and constructive comments on my manuscript entitled "Enterprise financialization and technological innovation: An Empirical Study Based on A-share Listed Companies Quoted on Shanghai and Shenzhen Stock Exchange". (ID: fintech-2260259). We have carefully considered the reviewers' suggestions and made some changes.

Every effort has been made to improve and make changes to the manuscript. These changes do not affect the content and framework of the paper. Yellow section modified according to your comments. We add to the research literature related to firm financialization and technological innovation.

Thank you so much for your attention and time. Looking forward to hearing from you.

your sincerity,

Corresponding Author:

Name: Sun Xinyu

Email: sxy17839973472@163.com